# ConCuR: Conciseness Makes State-of-the-Art Kernel Generation

## Abstract

GPU kernel generation by LLMs has recently experienced rapid development, leveraging test-time scaling and reinforcement learning techniques. However, a key challenge for kernel generation is the scarcity of high-quality data, as most high-quality kernels are proprietary and not open-source. This challenge prevents us from leveraging supervised fine-tuning to align LLMs to the kernel generation task. To address this challenge, we develop a pipeline that generates and curates high-quality CUDA kernels with reasoning traces, motivated by a critical observation that concise yet informative reasoning traces result in robust generation of high-performance kernels. Using this pipeline, we construct our dataset **ConCuR** and introduce our model **KernelCoder**, which is the first model trained on a curated dataset consisting of PyTorch, reasoning, and CUDA kernel pairs, to our knowledge. In the KernelBench setup, our model achieves significant improvements over the existing top-performing model, QwQ-32B, and outperforms all open-source models fine-tuned for kernel generation, as well as frontier models such as DeepSeek-V3.1-Think and Claude-4-sonnet. Finally, we show that the average reasoning length can serve as a metric to assess the difficulty of kernel generation tasks. The observations, metrics, and our data collection and curation pipeline can help obtain better data in the kernel generation task in the future.

## 1 Introduction

High-performance GPU kernels are critical to high performance in modern machine learning systems (Dao et al., 2022). However, developing them remains a costly and time-consuming task, demanding knowledge of domain-specific programming languages such as CUDA (Nickolls et al., 2008), Triton (Tillet et al., 2019), and ThunderKittens (Spector et al., 2025), as well as expertise in computer architectures. Therefore, tools are emerging to help developers develop GPU kernels. Previously, compilers like TVM (Chen et al., 2018) were introduced to generate program kernels. Recently, more works have focused on leveraging Large Language Models (LLMs) to generate CUDA or Triton kernels (Ouyang et al., 2025; Fisches et al., 2025). Initial attempts utilize test-time scaling (Terry Chen & Devleker, 2025; Lange et al., 2025), but these approaches are limited by the capabilities of the base models. Consequently, post-training techniques, especially reinforcement learning (RL), have been utilized to enhance the capabilities of models in the kernel generation domain (Baronio et al., 2025; Li et al., 2025b). Nevertheless, RL alone is often insufficient. It is observed that Supervised Fine-Tuning (SFT) on high-quality data appears indispensable, as SFT can provide foundational alignments for specific tasks. However, due to the scarcity of high-quality open-source CUDA kernels, we lack high-quality data to perform SFT on base models. This leads to the question: *How can we collect high-quality CUDA kernels to fully leverage the SFT method?*

In this work, we argue that a concise yet informative reasoning trace is crucial for generating high-quality CUDA kernels, since we observe that more outstanding generated CUDA kernels are accompanied by more concise reasoning traces. Based on this idea, we can select CUDA kernels paired with high-quality reasoning traces to construct a representative, informative, and well-curated dataset to enable effective SFT. Specifically, we introduce our data synthesis and curation pipeline (Figure 1), the first pipeline to select exceptional reasoning traces and excellent CUDA kernels in the kernel generation task. Our pipeline consists of two parts: data synthesis and data curation. For the data synthesis part, due to the scarcity of high-quality, open-source CUDA kernels, we choose to leverage existing reasoning models to synthesize CUDA kernels along with reasoning traces. For

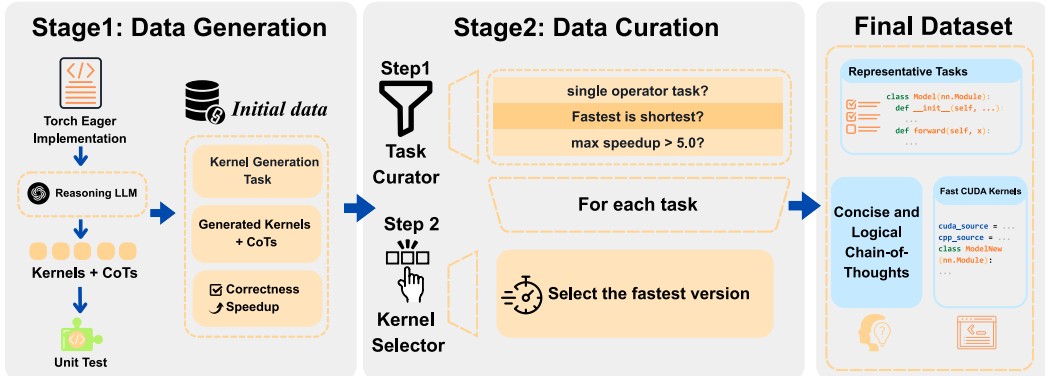

Figure 1: **Overview of our two-stage data gathering pipeline.** The first stage involves synthesizing CUDA kernels with corresponding CoTs and performing unit tests on each generated kernel to verify the correctness of the kernel and get the speedup over the torch eager implementation. The second stage involves selecting high-quality reasoning traces using our proposed criteria, based on the considerations of speedup, reasoning quality, and task distribution.

the data curation part, we design a data curation method that incorporates the conciseness of the reasoning trace and the performance of the kernel, motivated by our observation and idea. Based on this pipeline, we introduce the **Con**cise **CUDA R**easoning Dataset (**ConCuR**), a collection of 4,892 CUDA kernels paired with reasoning traces synthesized by Kevin-32B (Baronio et al., 2025). By performing LoRA (Hu et al., 2022) fine-tuning on QwQ-32B (QwenTeam, 2025) with our ConCuR dataset, we introduce **KernelCoder**, a state-of-the-art (SoTA) model on the kernel generation task.

Extensive experiments validate the high quality of our dataset and show that SFT remains an effective method for enhancing the model's ability to generate high-performance kernels. Specifically, we conduct comparisons with baselines via KernelBench (Ouyang et al., 2025) and show that KernelCoder outperforms all existing kernel generation models, as well as most frontier LLMs, such as GPT-4 (Hurst et al., 2024), DeepSeek-V3.1-Think (Liu et al., 2024), and Claude-4-Sonnet (Anthropic, 2025). Moreover, ablation studies prove that jointly incorporating the conciseness and performance of a generation (including a reasoning trace and a kernel), as well as balancing the type of generation tasks, are key to selecting representative tasks, high-quality reasoning traces, and CUDA kernels.

Additionally, based on extensive experiments, we find that the reasoning length could serve as an indicator of the complexity of kernel generation tasks. With this indicator, we can classify tasks into different difficulty levels, thereby creating a more rigorous benchmark for evaluating model performance. Furthermore, future studies can utilize this observation to select challenging kernel generation tasks, helping to construct more representative and rigorous datasets, which ultimately lead to the development of more powerful models.

In summary, our contributions are as follows:

1. **We argue that conciseness and informative reasoning trace results in a well-performed generated kernel.** We observe that concise reasoning traces lead to reliable and robust kernel generation. This argument contributes to the challenge of the lack of high-quality data in the kernel generation area.

2. **We propose a data synthesis and curation pipeline to build a high-quality dataset for the kernel generation task**. Motivated by our argument, we carefully design a data synthesis and curation pipeline. Utilizing this pipeline, we construct the first synthesized dataset of CUDA kernels with reasoning traces, ConCuR.

3. **Trained on ConCuR dataset, we introduce KernelCoder, a SoTA model capable of generating correct and efficient CUDA kernels**. Our model outperforms the frontier models and other competitors with fewer parameters and lower training cost.

4. **Validated by experiments, we demonstrate that reasoning length can be a suitable indicator to assess the difficulty of kernel generation tasks**. This indicator can help with data collection and model evaluation in future work.

## 2 RELATED WORK

### 2.1 GPU KERNEL OPTIMIZATION USING LLM

LLM benefits from scaling (Kaplan et al., 2020). To support models with an expanding number of parameters and larger-scale training, efficient GPU kernels are required. However, designing kernels requires a significant amount of time and resources, even for a team of experts. To accelerate this process, compilers such as TVM (Chen et al., 2018) and Taco (Kjolstad et al., 2017) have been developed. However, the ability of compilers is strong but limited compared to human experts. Recently, generating GPU kernels using LLM has gained wide attention. Evaluation benchmarks, such as KernelBench (Ouyang et al., 2025) and TritonBench (Li et al., 2025a), are constructed to facilitate systematic assessment of models' ability to design kernels in terms of accuracy and performance. Afterwards, test scaling, focusing on feedback-driven iterative approaches, has rapidly developed. A Nvidia team proposed a method to generate and refine kernels utilizing DeepSeek-R1 (Terry Chen & Develeker, 2025). The agent proposed by METR (METR, 2025) uses parallel tree search with verification to scale up. Orthogonal to these two works, the AI CUDA Engineer (Lange et al., 2025) attempted to utilize RAG (Gao et al., 2023) to scale test-time by increasing in-context learning.

Since the performance of frontier models on kernel generation falls short, more work has started to focus on training a more powerful model. KernelLLM (Fisches et al., 2025) constructed their dataset utilizing the Triton compiler to perform SFT. Kevin (Baronio et al., 2025), leveraging powerful reinforcement learning techniques, designed a multi-turn GRPO (Shao et al., 2024) training strategy and outperformed most of the frontier models. AutoTriton (Li et al., 2025b) employs both SFT and GRPO in its design. It constructed its SFT training dataset using LLM distillation and compilation with LLM-enhanced refinement simultaneously. More recently, the first multi-agent kernel generation system, Astra (Wei et al., 2025), was proposed. However, none of these works constructed a well-curated, high-quality dataset.

### 2.2 REASONING DATASET COLLECTION AND CURATION

The Chain-of-Thoughts (CoTs) significantly improves the ability of large language models to perform complex reasoning (Wei et al., 2022). Therefore, researchers have devoted efforts to constructing a high-quality dataset with labeled answers and CoTs to train reasoning models. LIMA (Zhou et al., 2023) demonstrated that a small set of high-quality data can achieve alignment for specific tasks. However, it is difficult to collect reliable data with chain-of-thoughts, especially for math and coding problems. Thus, employing LLMs to generate data appears to be a reasonable choice. WizardLM (Xu et al., 2025) proposed an iterative approach to generate complex science instructions along with their answers. Furthermore, s1 (Muennighoff et al., 2025) proposed a reasoning dataset curation method based on quality, difficulty, and diversity criteria.

## 3 CONCUR: SYSTEMATIC KERNEL AND COT CURATION

### 3.1 TASK DESCRIPTION

CUDA kernel design has been a challenging and time-consuming task, even for human experts. It requires both system-level hardware knowledge and algorithmic insight, including GPU architecture, parallel programming, algorithm design and memory optimization. The CUDA kernel generation task addresses this challenge by using a model to rewrite high-level PyTorch implementations into efficient, low-level CUDA kernels. It operates on PyTorch code with fixed input and hidden dimensions, where the scope ranges from single operators to entire networks. By replacing the complex and error-prone manual design, this automated generation hugely boosts developers' productivity.

## 3.2 KERNEL EVALUATION METRICS

To build a high-quality dataset, we must evaluate the kernels and select the high-performant ones. Following KernelBench (Ouyang et al., 2025), we evaluated kernels on two important metrics: **correctness** and **performance**. To evaluate the correctness, a random input is given to both the PyTorch implementation and the generated kernel. If the two outputs have the same dimension and values, the kernel is considered correct. To evaluate the performance of kernels, we choose speedup over the PyTorch Eager implementation as our metric:

$$\text{speedup} = \frac{T_{\text{Torch}}}{T_{\text{kernel}}} \cdot \mathbf{1}_{\text{correct}}[\text{kernel}], \tag{1}$$

where $T_{\text{Torch}}$ is the execution time of the PyTorch Eager implementation, $T_{\text{kernel}}$ is the execution time of the generated kernel, and $\mathbf{1}_{\text{correct}}[\cdot]$ is the indicator function for correct kernels.

## 3.3 REASONING DATA COLLECTION

We collected a total of 90,810 CUDA kernels with their corresponding CoTs. We selected the PyTorch programs from KernelBook (Paliskara & Saroufim, 2025) as our initial tasks. We generated five batches of kernels using Kevin-32B (Baronio et al., 2025) on 18,162 tasks and obtained 90,810 sample pairs (PyTorch program, CUDA kernel accompanied by CoT). Among them, 9,789 tasks yielded at least one correct kernel generation, producing a combined 24,136 correct kernels. While these kernels, along with their CoTs, are functionally correct, they are not of equal quality. To select the high-performant kernels, we therefore assess their quality based on the following two criteria:

1. **The speedup over PyTorch Eager (we refer to this as "speedup")**: as defined in Section 3.2, it tests if the generated kernel is efficient. A larger speedup indicates a higher efficiency.

2. **The length of the corresponding CoT in tokens (we refer to this as "reasoning length")**: it assesses how much computational resource the model invested to generate the kernel. For high-quality kernels, we anticipate their CoTs to be concise and logical.

From our experiments, we observed an unexpected correlation between these two criteria, which forms the foundation of our data curation method for a high-quality kernel dataset.

## 3.4 OBSERVATIONS ON GENERATION QUALITY AND REASONING LENGTH

In our first round of kernel generation, we identified two key observations regarding the relationship between reasoning length and the correctness and performance of the generated kernels on current LLMs.

1. **Shorter reasoning lengths are strongly associated with correct kernels (Figure 3)**. This observation contradicts previous opinions. The Deepseek-R1 paper (Guo et al., 2025) interprets an increase in reasoning length as a sign of an improved problem-solving ability of the model. The method of s1 (Muennighoff et al., 2025) assumes that a more challenging task requires more thinking tokens; thereby, they claim that a correct generation with a long reasoning trace would be high-quality data to learn. However, our observation illustrated that **although more challenging tasks typically require a greater number of reasoning tokens, for the same task, CUDA kernels generated after shorter reasoning traces tend to be correct more frequently than those produced through longer reasoning traces.**

Our detailed analyses (see Appendix B) of reasoning traces suggest the reason for this rela-

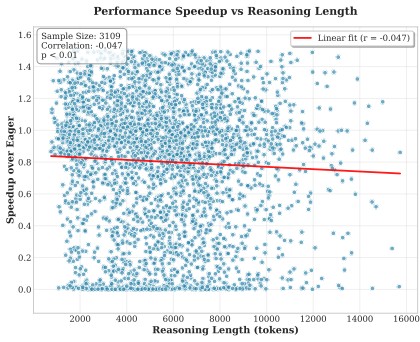

Figure 2: **Scatter plot showing the relationship between reasoning length (tokens) and speedup over eager execution.** A linear fit yields a correlation of $r = -0.047$ (Pearson correlation coefficient $p < 0.01$), indicating that reasoning length has virtually no practical impact on performance.

tionship: long reasoning demonstrates features
of overthinking (Chen et al., 2025; Wu et al., 2025).

In particular, lengthy reasoning often involves self-doubt and repeatedly verifies results that are already correct, which undermines logical coherence. In contrast, concise reasoning tends to be more logical and consistent, resulting in higher accuracy. In conclusion, we argue that a correctly generated kernel, along with a concise reasoning trace, is of high quality.

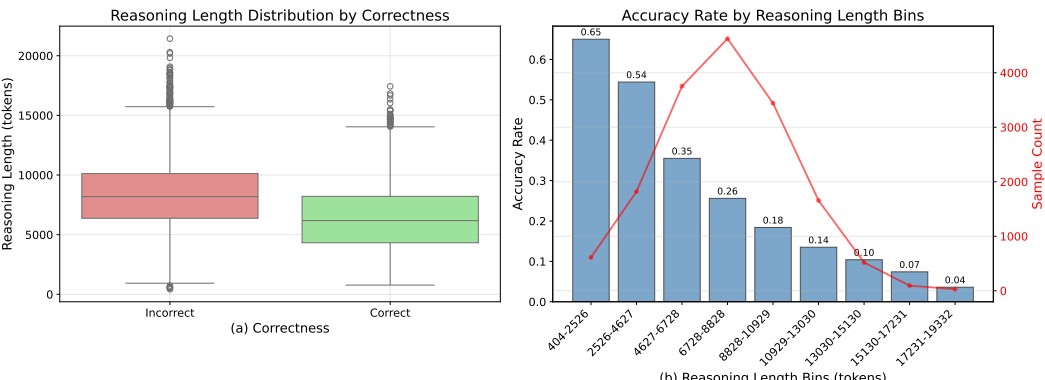

Figure 3: **Relationship between reasoning length and accuracy.** (a): Boxplot of reasoning length distributions for correct and incorrect responses, indicating that incorrect responses generally involve longer reasoning. (b): Accuracy (the proportion of correct generations among all generations) across reasoning length bins (blue bars) with corresponding sample counts (red line). The results indicate that shorter reasoning is generally associated with higher accuracy, whereas longer reasoning tends to reduce accuracy.

2. **We observed that speedup is largely independent of reasoning length (Figure 2).** From a conventional standpoint, longer reasoning processes, entailing more time spent on reasoning, are generally expected to enhance model performance. However, in CUDA kernel generation, we find that prolonged reasoning does not necessarily yield higher-quality kernels and may instead introduce redundant steps that neither enhance correctness nor improve execution efficiency. In fact, when the prompt does not explicitly specify the programming strategy, the model tends to generate similar high-level kernel optimization ideas and designs across trials. Consequently, extended reasoning merely adds redundant steps about these high-level ideas rather than improving the actual low-level implementation. This distinction is critical because it is these small variations in the low-level implementation—not the high-level idea—that ultimately dictate the speedup. Therefore, a concise reasoning chain that identifies the correct high-level idea can produce kernels just as efficient as, or even more so than, a longer and more verbose reasoning chain.

### 3.5 DATA CURATION

Based on our observations, we propose a data curation method to select high-quality CUDA kernels and reasoning CoTs, as shown in Figure 1. Our dataset, ConCuR, consists of three parts. (a) For each task, we generated five kernels. If the kernel with the shortest reasoning length achieves the highest speedup, we add it to our dataset. This approach prioritizes samples that simultaneously possess a short CoT and achieve a high speedup, as our observations indicate that shorter CoTs tend to be more logical and valuable. In total, we have 3,934 samples in this part. (b) We included kernels with speedups greater than 5, as these high-performance kernels are valuable for learning. This part contains 414 samples. (c) In addition, we found that designing a CUDA kernel for a single operator typically requires implementing it as a standalone kernel and optimizing it at the level of its individual operations. In contrast, designing kernels for multiple operators or entire networks primarily involves operator fusion. These represent two distinct design paradigms. Therefore, we need to balance the ratio of these two types of tasks in our dataset. To this end, we identified 544 samples with CUDA kernels for single operators and their CoTs. Based on these three parts, ConCuR, comprising 4,892 examples, considers speedup, reasoning quality, and task distribution.

Table 1: **Pass@1 results on KernelBench Level 1 and Level 2.** We present the execution accuracy (Exec) and the fast$_1$ score across levels, reported as percentages. The best result is labeled by **Bold**.

| Model | #Params | Lang. | Level 1 | | Level 2 | |
|---|---|---|---|---|---|---|
| | | | Exec↑ | fast$_1$↑ | Exec↑ | fast$_1$↑ |
| **Frontier Models** | | | | | | |
| DeepSeek-R1-0528 | 685B | Triton | 35.0 | 7.0 | 42.0 | 28.0 |
| DeepSeek-R1-0528 | 685B | CUDA | 52.0 | **18.0** | 55.0 | 38.0 |
| DeepSeek-V3.1-Think | 685B | CUDA | 44.0 | 16.0 | 30.0 | 20.0 |
| Qwen3-Coder-Plus | 480B | CUDA | 32.0 | 7.0 | 55.0 | 35.0 |
| GPT-4o | - | Triton | 15.0 | 3.0 | 5.0 | 3.0 |
| Claude-4-Sonnet | - | Triton | 33.0 | 11.0 | 26.0 | 10.0 |
| **Smaller-Scale Models** | | | | | | |
| QwQ | 32B | CUDA | 18.0 | 7.0 | 17.0 | 11.0 |
| Qwen3 | 8B | CUDA | 16.0 | 4.0 | 15.0 | 7.0 |
| Qwen3 | 32B | CUDA | 20.0 | 6.0 | 35.0 | 18.0 |
| Llama-3.3 | 70B | CUDA | 11.0 | 2.0 | 0.0 | 0.0 |
| AutoTriton | 8B | Triton | 36.0 | 10.0 | 45.0 | 17.0 |
| KernelLLM | 8B | Triton | 20.2 | - | 16.0 | - |
| Kevin* | 32B | CUDA | 50.0 | 16.0 | 46.0 | 27.0 |
| **KernelCoder** | 32B | CUDA | **58.0** | 17.0 | **59.0** | **39.0** |

In Section 5, we show that combining these three parts is crucial, as relying on one specific criterion or having an unbalanced task distribution will lead to a worse model.

## 4 EXPERIMENTAL RESULTS

### 4.1 TRAINING DETAILS

We select QwQ-32B (QwenTeam, 2025), a powerful reasoning model that excels at coding, as our base model. We leverage the ms-swift (Zhao et al., 2025) framework to perform LoRA (Hu et al., 2022) fine-tuning with ConCuR. We use a basic LoRA fine-tuning recipe, setting the LoRA rank to 32, the LoRA alpha to 32, and the LoRA dropout to 0.05 for all linear layers. The model is trained for three epochs with a batch size of 8 and a gradient accumulation of 2, resulting in a total of 1,815 gradient steps. The learning rate is set to 1e-4 with a warmup ratio of 0.05, and then decays following a cosine schedule. We use the AdamW (Loshchilov & Hutter, 2019) optimizer with $\beta_1 = 0.9$, $\beta_2 = 0.95$, and weight decay of $0.1$. We do not compute loss on system prompts, only on responses (CoTs and generated kernels). Due to the long reasoning trace in our dataset, we set the max sequence length to 32,768. We use ZeRO stage-3 (Rajbhandari et al., 2020) to train our model on a single node with 8 A100 GPUs. The training process takes 9 hours, requiring significantly less computational resources than its counterparts. Our training dynamics are reported in Appendix A.

### 4.2 EVALUATION

We evaluated the models on KernelBench (Ouyang et al., 2025), which consists of four levels with increasing difficulties. We select level 1 (single-kernel operators, such as matrix multiplication and convolution) and level 2 (simple fusion patterns, such as conv + bias + relu) as our evaluation set. Both level 3 and 4 are challenging and exceed the capabilities of current LLMs to generate meaningful kernels, so they are not included in our evaluation set. We adopt two metrics to evaluate the model on each level: (1) **Exec**, the fraction of correct kernels (Li et al., 2025b) and (2) **fast$_p$**, the fraction of kernels that are both correct and have a speedup greater than a threshold $p$ (Ouyang et al., 2025):

$$\text{fast}_p = \frac{1}{N} \sum_{i=1}^{N} \mathbf{1}_{\text{correct}}[\text{speedup}_i > p], \tag{2}$$

Table 2: **Pass@10 results on KernelBench Level 1 and Level 2.** The results show that our model is capable of solving most tasks in KernelBench successfully by simply employing parallel inference. DeepSeek-V3.1-Think performs worse than DeepSeek-R1-0528 since the CoTs of V3.1 are highly compressed. This compression decreases the quality of CoTs. In contrast, our method, which prefers short CoTs, successfully selects concise but high-quality CoTs.

| Model | #Params | Lang. | Level 1 | | Level 2 | |
|---|---|---|---|---|---|---|
| | | | Exec↑ | fast$_1$↑ | Exec↑ | fast$_1$↑ |
| **Frontier Models** | | | | | | |
| AI CUDA Engineer | | | | | | |
|   -o1-preview | - | CUDA | 63.0 | - | 95.0 | - |
|   -o1-high | - | CUDA | 50.0 | - | 81.0 | - |
| DeepSeek-R1-0528 | 685B | CUDA | 90.0 | 31.0 | **97.0** | **82.0** |
| DeepSeek-V3.1-Think | 685B | CUDA | 72.0 | 24.0 | 78.0 | 61.0 |
| Qwen3-Coder-Plus | 480B | CUDA | 76.0 | **35.0** | 94.0 | 76.0 |
| Claude-4-Sonnet | - | CUDA | 64.0 | - | 92.0 | - |
| **Smaller-Scale Models** | | | | | | |
| QwQ | 32B | CUDA | 55.0 | 12.0 | 76.0 | 46.0 |
| Qwen3 | 8B | CUDA | 31.0 | 12.0 | 53.0 | 32.0 |
| Qwen3 | 32B | CUDA | 68.0 | 21.0 | 82.0 | 37.0 |
| Llama-3.3 | 70B | CUDA | 31.0 | 9.0 | 8.0 | 1.0 |
| KernelLLM | 8B | Triton | 52.0 | - | 34.0 | - |
| AutoTriton | 8B | Triton | 68.0 | - | 88.0 | - |
| Kevin* | 32B | CUDA | 86.0 | 20.0 | 90.0 | 63.0 |
| **KernelCoder** | 32B | CUDA | **91.0** | 32.0 | 95.0 | 68.0 |

where $N$ is the number of tasks at this level.

We compare our model against frontier models and open-source fine-tuned models. We report pass@1 and pass@10 (Du et al., 2024) results here. For the Exec score, pass@10 checks if there is at least one correct kernel among 10 trials. For the fast$_1$ score, pass@10 checks if there is at least one kernel with a speedup larger than 1. As shown in Table 1 and Table 2, our model has made a significant leap in correctness (Exec) and performance (fast$_1$) compared to our base model, QwQ-32B. Moreover, it surpasses all frontier models, including DeepSeek-R1-0528, GPT-4o, and Claude-4-sonnet, as well as fine-tuned models like Kevin, especially in generating correct kernels, demonstrating that our dataset is of high quality. Experimental results show that our data curation method distills concise and logical reasoning traces, equipping our model with robust and error-resistant reasoning patterns. All evaluations are run on a node with 8 RTX 5090 GPUs.

## 4.3 EFFICIENCY

We compare our model against frontier models and open-source fine-tuned models in terms of the efficiency of training samples and computational resources. As shown in Table 3, KernelCoder is trained on fewer training samples with a lower cost of computational resources. It only takes 4,892 samples and 64 A100 GPU hours, while the second best, Kevin, takes more than 600 H200 GPU hours, highlighting the efficiency of KernelCoder.

# 5 ABLATION STUDY

## 5.1 DATA CURATION

In this section, we prove that combining the two criteria stated in Section 3.3 and balancing the types of tasks are crucial for selecting high-quality reasoning traces and CUDA kernels. We construct the following datasets and train on them using the same settings as in the main experiments.

Table 3: **Comparison on efficiency of training samples and computational resources**. We present the size of parameters of models (#param), the post-training methodology (method), the number of training samples (#samples), the computational resources cost in terms of GPU hours (resource) on A100/H100, and pass@10 results on KernelBench level 1 and 2 (performance). *: Kevin used 180 problems of KernelBench. For each problem, it explores 16 trajectories in parallel with 8 refinement steps.

| Model | #Params | Method | #Samples | Resource | Performance |
|---|---|---|---|---|---|
| DeepSeek-R1-0528 | 685B | - | - | - | 90/97 |
| AutoTriton | 8B | SFT+GRPO | 14102+6302 | 128+512 | 68/88 |
| KernelLLM | 8B | SFT | 25000 | 192 | 52/34 |
| Kevin* | 32B | GRPO | 180* | $> 600$ | 86/90 |
| **KernelCoder** | 32B | SFT | 4892 | 64 | 91/95 |

1. **Random Selection (5K-random)**: For each task, we randomly select one correct kernel with its CoT (if such a kernel does not exist, we skip). Then, we randomly select 4,892 tasks along with their corresponding kernel.

2. **Max Length First Selection (5K-max)**: For each task, we select the correct kernel with the longest reasoning length, and then retain the 4,892 tasks whose kernels exhibit the longest reasoning overall. This method was introduced by s1 (Muennighoff et al., 2025), which hypothesized that a more challenging task would require more reasoning tokens.

3. **Min length First Selection (5K-min)**: For each task, we select the correct kernel with the shortest reasoning length, and then retain the 4,892 tasks whose kernels exhibit the shortest reasoning overall. This method only considers the conciseness (criterion 1) of the reasoning trace and would potentially select only easy tasks.

4. **Speedup First Selection (5K-speedup)**: We construct the dataset by selecting, for each task, the kernel with the highest speedup and then keeping the 4,892 tasks with the largest speedups overall. This method only considers the performance of kernels (criterion 2). Although models can learn from high-performance kernels, this method may select only easy-to-optimize tasks, resulting in the model struggling with more challenging tasks.

Table 4: **Ablation study on data selection methods**. The results are evaluated for both pass@1 (shown on the left) and pass@10 (shown on the right). ARL denotes the Average Reasoning Length (in tokens) for the generated CoTs at each level.

| Model | Level 1 | | | Level 2 | | |
|---|---|---|---|---|---|---|
| | Exec↑ | fast$_1$↑ | ARL | Exec↑ | fast$_1$↑ | ARL |
| 5K-random | 39.0 / 84.0 | 9.0 / 21.0 | 7065.3 | 50.0 / 90.0 | 24.0 / 55.0 | 6447.2 |
| 5K-max | 34.0 / 86.0 | 7.0 / 17.0 | 7238.4 | 53.0 / 96.0 | 27.0 / 50.0 | 6515.3 |
| 5K-min | 35.0 / 86.0 | 10.0 / 23.0 | 6710.9 | 50.0 / 91.0 | 26.0 / 55.0 | 6100.7 |
| 5K-speedup | 42.0 / 83.0 | 8.0 / 21.0 | 7119.3 | 52.0 / 93.0 | 21.0 / 56.0 | 6435.0 |
| **KernelCoder** | 58.0 / 91.0 | 17.0 / 32.0 | 7035.9 | 59.0 / 95.0 | 39.0 / 68.0 | 6410.8 |

As shown in Table 4, considering only one criterion would cause worse model performance, especially in correctness. Notably, KernelCoder substantially outperforms other models, which can be attributed to its higher single-attempt correctness: although other models can approach Kernel-Coder's correctness over multiple attempts, the consistently strong performance in a single attempt demonstrates that KernelCoder is less prone to errors. Trained on these datasets, models will learn similar patterns, methods, ideas, and techniques to design kernels. However, the quality of the reasoning process leads to different abilities for the models to reason effectively and correctly, which is crucial to generating correct kernels. Besides, these four datasets we construct for the ablation study do not balance the types of tasks. Therefore, models trained on these datasets have worse performances than KernelCoder on KernelBench Level 1. 5K-max, which utilizes the method proposed by

s1 (Muennighoff et al., 2025), has improved the performance on KernelBench Level 2, but overall still fails to outperform our model.

Additionally, we analyze the average reasoning length (ARL) of each model on levels, defined as:

$$\text{ARL} = \frac{1}{NM} \sum_{i=1}^{N} \sum_{j=1}^{M} L[i,j], \tag{3}$$

where $L \in \mathbb{N}^{N \times M}$ is the reasoning length of all the samples. $N$ is the number of tasks in this level, and $M$ is the number of times each task is generated. The ARL results demonstrate that relying solely on one criterion would cause bias in the dataset. 5K-max has selected some long but illogical, chaotic traces, so its ARL is longer than 5K-random and KernelCoder. 5K-min would mistakenly have a bias towards simple tasks. Consequently, it may not invest enough reasoning tokens to solve hard tasks. However, ConCuR has balanced and unbiased data, as the ARL of KernelCoder is close to that of 5K-random, which potentially approaches the optimal reasoning length (Wu et al., 2025).

## 5.2 BASE MODEL

In this section, we demonstrate that our findings are not tied to a specific base model. We fine-tune three different base models on ConCuR. As shown in Table 5, all fine-tuned models outperform their respective base models. This result further strengthens the empirical evidence and highlights the broader applicability of our ConCuR dataset.

Table 5: **Pass@10 results of different base models and fine-tuned models**.

| Model | Level 1 | | Level 2 | |
|---|---|---|---|---|
| | Exec↑ | fast$_1$↑ | Exec↑ | fast$_1$↑ |
| Qwen3-8B | 31.0 | 12.0 | 53.0 | 32.0 |
| **Qwen3-8B-SFT** | 47.0 | 10.0 | 89.0 | 58.0 |
| Qwen3-32B | 68.0 | 21.0 | 82.0 | 37.0 |
| **Qwen3-32B-SFT** | 72.0 | 21.0 | 94.0 | 68.0 |
| QwQ-32B | 55.0 | 12.0 | 76.0 | 46.0 |
| **KernelCoder** | **91.0** | 32.0 | 95.0 | 68.0 |

## 6 DISCUSSION

### 6.1 DIVISION OF TASK DIFFICULTY

Currently, there is no suitable criterion to assess the difficulty of one kernel generation task, which is crucial for constructing high-quality datasets and benchmarks. KernelBench (Ouyang et al., 2025) utilizes the model's structure to categorize tasks into multiple levels. However, as shown in Table 2, all models perform worse on level 1 than on level 2, indicating that some tasks in level 1 are more challenging to solve than some in level 2 (a notable example is convolution). To assess the inherent difficulty of kernel generation tasks, we propose a method that compares task difficulty based on ARL. For a given set of tasks, we first select a sufficiently strong reasoning model (e.g., Kevin-32B or DeepSeek-R1-0528) and generate $M$ times for each task. We then compute the ARL for each task over $M$ generations. Since individual reasoning processes exhibit variability, we perform multiple generations for each task to mitigate bias. As $M$ increases, the ARL becomes a more reliable estimator of the inherent difficulty of the task.

### 6.2 DIFFICULTY DIVISION OF KERNELBENCH

In our experiment, we use Kevin as the generator and selected $M = 10$. We categorize KernelBench level 1 and level 2 into three difficulty levels: easy, medium, and hard, based on the ARL. The thresholds and task counts are listed in Table 6. As shown in Table 7, across most models, both

Table 6: **Difficulty division of KernelBench Level 1 and Level 2.** Tasks are divided into three difficulty levels based on their ARLs.

| Level | ARL | Number |
|---|---|---|
| Easy | $< 4000$ | 37 |
| Medium | $4000 \sim 8500$ | 114 |
| Hard | $> 8500$ | 49 |

accuracy and performance consistently decrease from the easy subset to the hard subset. DeepSeek-R1-0528, although the best-performing model overall, fails to achieve a perfect score on the Easy level. This gap may stem from the inherent ability difference between Kevin and DeepSeek-R1-0528. Overall, this trend indicates that tasks are successfully divided by level of difficulty. This task division method can facilitate the construction of more challenging benchmarks and datasets.

Table 7: **Pass@10 results of different models on KernelBench difficulty division.** The performance is reported as the geometric average of speedups ($G_{speedup}$).

| Models | Easy | | Medium | | Hard | |
|---|---|---|---|---|---|---|
| | Exec$\uparrow$ | $G_{speedup}\uparrow$ | Exec$\uparrow$ | $G_{speedup}\uparrow$ | Exec$\uparrow$ | $G_{speedup}\uparrow$ |
| Kevin-32B | 100.0 | 1.215 | 91.2 | 0.752 | 67.3 | 0.376 |
| Qwen3-8B | 83.8 | 1.229 | 40.4 | 0.428 | 14.3 | 0.675 |
| DeepSeek-V3.1-Think | 91.9 | 1.218 | 80.7 | 0.747 | 49.0 | 0.399 |
| DeepSeek-R1-0528 | 94.6 | 1.869 | 95.6 | 2.515 | 87.8 | 1.276 |
| Qwen3-Coder-Plus | 100.0 | 1.468 | 86.8 | 1.152 | 69.4 | 0.741 |
| KernelCoder | 100.0 | 1.319 | 94.7 | 0.831 | 83.7 | 0.410 |

## 7 CONCLUSION

### 7.1 SUMMARY

We design a data collection and curation pipeline to address the lack of high-quality data in the kernel generation area, based on our insight into the length of the reasoning and the performance of the generated kernel. Utilizing this pipeline, we construct the dataset ConCuR, the first curated dataset of CUDA kernels with reasoning traces. We also present KernelCoder trained on ConCuR. Experiment results on KernelBench demonstrate that KernelCoder outperforms existing models fine-tuned on the kernel generation task, as well as frontier models, which indicates that our data collection and curation pipeline successfully selects high-quality data. By addressing the scarcity of high-quality data, our work demonstrates that SFT remains crucial for enhancing a model's kernel generation capability. Additionally, since both test-time scaling approaches and RL approaches require a powerful base model, our model can facilitate these approaches by providing a pathway to obtain a more powerful base model. Finally, we propose a method of dividing tasks by difficulty based on reasoning length, which also helps select more valuable tasks to construct better benchmarks and datasets in the future.

### 7.2 FUTURE WORK

Currently, models are capable of generating correct kernels for most tasks. However, the generated kernels do not exhibit satisfactory performance (at least, better than Torch Eager), which is a common phenomenon for all models. This can be attributed to the model rewriting the non-bottlenecking part into kernels, rather than optimizing the bottlenecking part. This can be potentially solved by developing a paradigm involving multi-agent work and test-time scaling, such as Astra (Wei et al., 2025), which includes profiling, idea generation, and kernel implementation. Based on this paradigm, more diverse, reasonable, and comprehensive datasets (in terms of patterns, ideas, and methods for designing kernels) could be constructed, and our data curation method could be applied to these datasets to select high-quality data.

## 8 REPRODUCIBILITY STATEMENT

Our curated dataset ConCuR can be reproduced by following these steps: 1. Use reasoning models to generate kernels with CoTs using the prompt provided in Appendix C. 2. Use the data curation method introduced in Section 3.5. To reproduce our model KernelCoder, please refer to the training details in Section 4.1.

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

## A TRAINING DETAILS

We report our training dynamics during LoRA fine-tuning and the results of other ablation studies here.

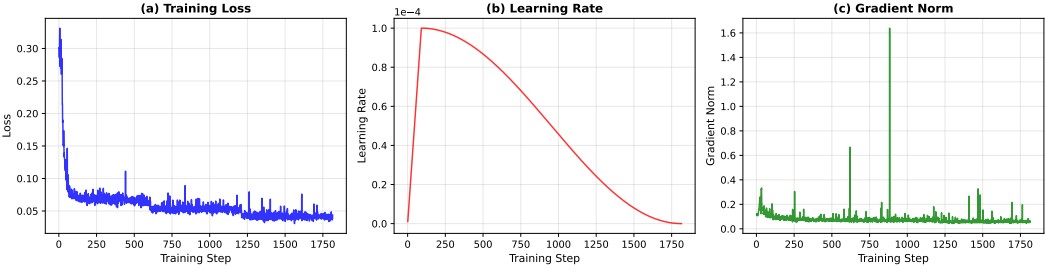

Figure 4: **Training statistics of KernelCoder on ConCuR.**

### A.1 TRAINING ABLATIONS: SIZE OF DATASET

Table 8: **Pass@10 results of models trained on datasets of different sizes.**

| #Data | Level1 | | Level2 | |
|---|---|---|---|---|
| | Exec | fast$_1$ | Exec | fast$_1$ |
| 552 | 72.0 | 21.0 | 84.0 | 50.0 |
| 1105 | 80.0 | 32.0 | 89.0 | 64.0 |
| 2210 | 86.0 | 26.0 | 94.0 | 66.0 |
| 4892 | 91.0 | 32.0 | 95.0 | 68.0 |

Besides the ablation study in Section 5, we also implement an ablation study on the size of our dataset. We randomly select #Data of samples from the ConCuR to construct three smaller datasets. The results shown in Table 8 demonstrate that increasing the dataset size improves the model's ability, but this improvement is subject to a margin effect decrease. Nevertheless, the size of 4,892 samples is significantly smaller than the size of the training dataset of AutoTriton (Li et al., 2025b) and KernelLLM (Fisches et al., 2025), which are 14,102 and approximately 25,000, respectively.

### A.2 TRAINING COST

As shown in Table 9, our method requires significantly fewer computational resources than its counterparts. For KernelLLM and AutoTriton, although the scale of parameters of our model is larger than theirs, our dataset is on a smaller scale. Moreover, we do not employ GRPO, which is an effective yet computationally heavy method. For Kevin, their multi-turn setup amplifies the computational cost of GRPO, which involves sampling $m$ parallel trajectories with $n$ refinement turns for each task. Therefore, the VRAM requirement and computational cost Kevin required are significantly higher than ours.

Table 9: **Comparison on training cost across existing fine-tuned models.** AutoTriton used 14,102 samples for 3 epochs for SFT and 6,302 samples for 1 epoch for GRPO. * denotes that Kevin used 180 evaluation data for training.

| Models | #Params | Methods | #Data | epochs |
|---|---|---|---|---|
| Kevin | 32B | GRPO (multi-turn) | 180* | - |
| KernelLLM | 8B | SFT | 25000 | 10 |
| AutoTriton | 8B | SFT + GRPO | 14102 + 6302 | 3 + 1 |
| KernelCoder | 32B | LoRA | 4892 | 3 |

# B    CASE STUDY

We present the reasoning trace generated by Kevin-32B (Baronio et al., 2025) in parallel inference for KernelBench Level 1 Problem 1 (square matrix multiplication). The generated kernel of the concise version is correct, while the kernel of the long version is incorrect. As shown in the following text boxes, the concise version has a clear initial plan, a coherent logical flow, and organized execution. However, the long version struggles with self-doubt from the beginning, giving rise to an incomplete plan that requires continuous revision and refinement. Therefore, it is more prone to error and culminates in an incorrect kernel generation. Additionally, we observed several times that the reasoning trace is wrong or of low quality from the outset, resulting in errors and disruptions in subsequent generations. Therefore, the method of iterative refinement sometimes proved ineffective, as the generated reasoning is not of good quality from the outset. To solve this problem, parallel inference can be employed to generate multiple reasoning traces simultaneously, allowing selection of the highest-quality trace for subsequent steps.

---

Concise Version (4814 tokens, correct kernel generated):
Okay, I need to optimize the given Model which performs a matrix multiplication using torch.matmul. The goal is to write a custom CUDA kernel to replace this operation for better performance. Let me think about how to approach this.
...
The steps are:
1. Each block loads a tile of A and a tile of B into shared memory.
2. Each thread computes its part of the dot product using the tiles.
3. Use multiple passes to handle larger matrices than what fits in shared memory.
Wait, but this is a very naive approach and not optimized. The better approach uses shared memory to store tiles of A and B, so that each thread can compute multiple elements. Let me think again.
A better approach is the matrix multiplication using tiling. Here's the standard approach:
Each block is responsible for a tile of the output matrix C. The block size is BLOCK_SIZE x BLOCK_SIZE (e.g., 16x16). Each thread in the block computes one element of the tile. The kernel uses shared memory to store the tiles of A and B.
The kernel would look something like:
...
Now, in the Python code, I need to compile this kernel. The Python code will have the CUDA source as a string, then load it using load_inline.
So the steps are:
1. Write the CUDA kernel code as a string.
2. Define the Python function that calls the kernel.
3. Create a new ModelNew class that uses this function.
Let me structure the code.
...
Putting it all together, the Python code would be:

---

Long Version (6575 tokens, wrong kernel generated):
Okay, I need to optimize the given Model which performs a matrix multiplication using torch.matmul. The user wants me to write a custom CUDA kernel to replace this operation for better performance. Let me think about how to approach this.
First, the original model just does a simple matrix multiply of two N x N matrices. The standard torch.matmul is already quite optimized, but maybe there's a way to make it faster with a custom kernel, especially for a specific matrix size like 2048*2. Perhaps by using shared memory to store chunks of the matrices and reducing memory access time?
Wait, but matrix multiplication is a highly optimized operation in cuBLAS, which is what PyTorch uses under the hood. So maybe the default implementation is already as fast as it can get. Hmm, but the problem says to try replacing it. Maybe the user wants to experiment with block matrix multiplication or some specific tiling approach?
Let me outline the steps:

---

1. Each block of threads is responsible for a tile of the output matrix C. The tile size should be chosen based on the shared memory available. Let's say a tile size of 16x16 or 32x32.
2. The kernel will have each block load a tile of A and a tile of B into shared memory. Then, compute the dot products for the corresponding tile in C.

...
But to reduce global memory access, the blocks of A and B are first loaded into shared memory. So each block will have two shared memory arrays: one for the current tile of A, and one for the current tile of B.
The process would be:
1. Each block is responsible for a block of C at (blockIdx.y, blockIdx.x) or something like that. Wait, the block index needs to map to the position of the C tile.
2. The kernel will have to loop over all the tiles of A and B that are needed to compute the C tile. For each iteration, load the respective tiles of A and B into shared memory, synchronize, compute the partial results, and then sum them up.

...
Wait, in the B loading part, ...
Ah, here's the mistake...
Wait, no. The kernel as written has each thread responsible for one element in the C block...
Wait, perhaps the sharedA is the block of A's rows (blockRow*BLOCK_SIZE to (blockRow+1)*BLOCK_SIZE) and columns m*BLOCK_SIZE to (m+1)*BLOCK_SIZE...
Now, the kernel launch parameters. The grid should be divided into blocks of size BLOCK_SIZE x BLOCK_SIZE. The number of blocks in x and y dimensions is ceil(N BLOCK_SIZE).
...
Putting it all together, the new code would be:

## C  INFERENCE PROMPT

```
PROMPT_TEMPLATE = Template('''
You are a Machine Learning Engineer trying to write custom cuda kernels
    to replace the pytorch operators in the given architecture to get
    speedups.
You have complete freedom to choose the set of operators you want to
    replace. You may make the decision to replace some operators with
    custom cuda kernels and leave others unchanged. You may replace
    multiple operators with custom implementations, consider operator
    fusion opportunities (combining multiple operators into a single
    kernel, for example, combining matmul+relu), or algorithmic changes (
    such as online softmax). You are only limited by your imagination.

For [Imports], you will likely need but not limited to the following
    libraries:
```
import torch
import torch.nn as nn
import torch.nn.functional as F
import math
```

Here's an example to show you the syntax of inline embedding custom
    operators from the cuda kernel in torch:
The pytorch module needed to be optimize is:
```
$ref_arch_torch
```

The example new arch with custom cuda kernels looks like this:
```
$ref_arch_kernel
```
```

```
And the PyTorch code you need to optimize is:
```
$code
```
Optimize the architecture named Model with custom cuda kernels! Optimize
    the architecture named Model with custom cuda kernels! Name your
    optimized output architecture ModelNew. Output the new code in
    codeblocks. Please generate real code, NOT pseudocode, make sure the
    code compiles and is fully functional. Just output the new model code
    , no other text, and NO testing code!
''')
```

## D   LLM USAGE

I used LLM to check my grammar and polish my writing. Besides, I also used LLM to help find some papers about dataset collection and curation.

