# OpenReview forum: "ConCuR: Conciseness Makes State-of-the-Art Kernel Generation"
_ICLR.cc/2026/Conference — Submitted to ICLR 2026_

### Official Review · Reviewer_2Vjz · 2025-10-28

**Soundness:** 3
**Presentation:** 3
**Contribution:** 3
**Rating:** 6
**Confidence:** 4

**Summary:**

The paper makes a solid and well-motivated contribution to kernel generation. It introduces the ConCuR dataset based on the novel observation that concise reasoning traces improve kernel generation, and presents KernelCoder, a state-of-the-art model capable of producing correct and efficient CUDA kernels. The work is clearly written, experimentally strong, and supported by thorough ablation studies.

**Strengths:**

The paper presents a substantive and well-motivated contribution to the challenging task of kernel generation.

1. The authors make an insightful observation that concise reasoning traces lead to better kernel generation performance. Building on this finding, they construct a new dataset (ConCuR) specifically designed around this principle.
2. The proposed model, KernelCoder, demonstrates strong technical quality. It is a state-of-the-art model capable of generating correct and efficient CUDA kernels.
3. The paper is clearly structured and well-presented. The motivation, methodology, and results are logically connected, making it easy to follow the authors’ reasoning and understand the contributions.
4. The experimental results show that KernelCoder consistently outperforms both frontier and fine-tuned models on KernelBench Level 1 and Level 2 benchmarks. The inclusion of an ablation study further strengthens the work by demonstrating the effectiveness of the dataset curation pipeline.

Overall, the paper offers novel insights and practical advancements for the field of kernel code generation. Through the creation of the ConCuR dataset and the development of KernelCoder, the authors provide a meaningful contribution that can inspire future research on reasoning-based code generation.

**Weaknesses:**

1. DeepSeek-R1 appears to be a very strong general-purpose reasoning model. When compared with KernelCoder, which is specifically trained for kernel generation, the performance gap does not seem particularly large.
2. The paper should provide more information about the *Correctness Analysis* setup — including how many random inputs were used for validation, and what tolerance thresholds were applied when judging correctness.
3. It is unclear why KernelBench Level 3 experiments were not included. A justification for this maybe helpful.
4. In Section 3.4, for the second part of the dataset, the authors select samples achieving 5× speedup. The rationale for this specific threshold is not explained. It would be interesting to know how results might change with alternative thresholds (e.g., 3× speedup).
5. It would be valuable to include the performance of DeepSeek-R1-0528 in Table 5 for a more complete comparison.
6. Why choose Kevin to create dataset. possible to use DeepSeek-R1-0528? Or rounded to use kernelCoder to create dataset for next round training.
7. The dataset was created using Kevin, but the motivation for choosing this model over others (e.g., DeepSeek-R1-0528) is not fully explained. And is it possible to use KernelCoder itself to create dataset for next round training.

**Questions:**

Please refer to the Weaknesses section.

---

> ### Author Response · Authors · 2025-11-22
>
> Response to Reviewer `2Vjz`: We thank you for your detailed and thoughtful feedback. Below, we address the key concerns raised in your review.
>
> ---
>
> > **W1:** *'DeepSeek-R1 appears to be a very strong general-purpose reasoning model. When compared with KernelCoder, which is specifically trained for kernel generation, the performance gap does not seem particularly large.'*
>
> **A:** Thank you for pointing this out. However, DeepSeek-R1 is substantially larger and trained on far more data than KernelCoder, which partially explains its strong performance despite not being specialized for kernel generation (please refer to Section 4.3 in revised version for detailed comparison). More importantly, KernelCoder is a lighter and faster model that can be easily served locally. As a result, it may be better suited for multi-agent frameworks, especially given that recent works indicate scaling through multi-agent setups is an effective approach to generate correct and high-performance kernels.
>
> > **W2:** *'The paper should provide more information about the Correctness Analysis setup — including how many random inputs were used for validation, and what tolerance thresholds were applied when judging correctness.'*
>
> **A:** We stick to KernelBench default configurations, where 100 randomly generated inputs are tested, and generated kernels are considered as correct if they pass all 100 tests. The threshold is 1e-4.
>
> > **W3:** *'It is unclear why KernelBench Level 3 experiments were not included.'*
>
> **A:** Thank you for this feedback. We have also evaluated KernelBench Level 3 (in fact, KernelCoder performs even better at this level). However, after a detailed analysis of the generated kernels, we found that most existing models optimize only a small portion of the overall model architecture at Level 3. Consequently, the level 3 results do not reliably reflect the true performance of the models. For this reason, we chose to report only the results for KernelBench Levels 1 and 2.
>
> > **W4:** *'In Section 3.4, for the second part of the dataset, the authors select samples achieving 5× speedup. The rationale for this specific threshold is not explained. It would be interesting to know how results might change with alternative thresholds'*
>
> **A:** Thank you for raising this point. We are happy to provide more explanation regarding the choice of the threshold. The motivation behind this subset of samples is to ensure that we do not discard exceptionally high-quality kernels, particularly those accompanied by long and informative reasoning traces. At the same time, we need to prevent the dataset size from growing excessively large. The choice of 5× serves as a balance between these two considerations: it preserves standout samples while keeping the dataset size manageable.
>
> To further validate this design choice, we trained additional models using alternative thresholds. The results are shown below:
>
> | Threshold| Level 1 Exec ↑ | Level 1 fast₁ ↑ | Level 2 Exec ↑ | Level 2 fast₁ ↑ |
> |---|---|---|---|---|
> |3|92.0 |32.0 |93.0 |63.0 |
> |7|90.0 |31.0 |92.0|70.0|
> |**KernelCoder (5)**   | 91.0| 32.0| 95.0| 68.0|
>
> Overall, the performance differences across thresholds are small. However, the variation in fast_1 suggests that the threshold does have some influence on the quality of optimized kernels. We believe the choice of 5 represents a reasonable middle ground, but we appreciate the suggestion and will explore a more systematic threshold selection strategy in future work.
>
> > **W5:** *'It would be valuable to include the performance of DeepSeek-R1-0528 in Table 5 for a more complete comparison.'*
>
> **A:** Thank you for this suggestion. We now include the new Table **7** in revised version. For your convenience, we list it here.
>
> | Models                | Easy Exec ↑ | Easy G_speedup ↑ | Medium Exec ↑ | Medium G_speedup ↑ | Hard Exec ↑ | Hard G_speedup ↑ |
> |----------------------|--------------|-------------------|----------------|----------------------|--------------|--------------------|
> | Kevin-32B            | 100.0        | 1.215             | 91.2          | 0.752                | 67.3        | 0.376              |
> | Qwen3-8B             | 83.8         | 1.229             | 40.4          | 0.428                | 14.3        | 0.675              |
> | DeepSeek-V3.1-Think  | 91.9         | 1.218             | 80.7          | 0.747                | 49.0        | 0.399              |
> | DeepSeek-R1-0528     | 94.6         | 1.869             | 95.6          | 2.515                | 87.8        | 1.276              |
> | Qwen3-Coder-Plus     | 100.0        | 1.468             | 86.8          | 1.152                | 69.4        | 0.741              |
> | KernelCoder          | 100.0        | 1.319             | 94.7          | 0.831                | 83.7        | 0.410              |

---

> ### Author Response · Authors · 2025-11-22
>
> > **W6&7:** *'Why choose Kevin to create dataset. possible to use DeepSeek-R1-0528? Or rounded to use kernelCoder to create dataset for next round training.'*
>
> **A:** We appreciate your concern regarding the choice of model used for dataset generation. Although employing DeepSeek-R1-0528 could potentially yield higher-quality samples, its cost is prohibitive. As a reference, generating five samples (CoT + kernel) for a single problem costs approximately `$0.175` (therefore `$3100` in total). For this reason, we opted to use Kevin, the previous state-of-the-art model that can be served locally for dataset generation.
>
> Furthermore, while using KernelCoder to create a dataset for the next round of training is a reasonable direction, we argue that reinforcement learning–based approaches or test-time scaling strategies may provide a more computationally efficient alternative. We leave a deeper exploration of these methods for future work.
>
> We appreciate the time and effort you’ve dedicated to reviewing our submission. Your thoughtful feedback has significantly strengthened our paper.

---

### Official Review · Reviewer_k3BA · 2025-10-29

**Soundness:** 2
**Presentation:** 2
**Contribution:** 1
**Rating:** 2
**Confidence:** 4

**Summary:**

Authors propose data generation and curation strategy for kernel generation using LLMs. Authors emphasize on generating chain-of-thought representation along with kernel code for supervised finetuning (SFT) of LLMs. Authors further demonstrate the efficacy of this strategy for SFT on opensource models. Authors point out two key and non-intuitive observations: 1) shorter reasoning lengths lead to better performance and 2) generated kernel performance improvement is not correlated to reasoning length. Authors compare their results with both open and closed source frontier models. KernelCoder, LLM finetuned with ConCuR strategy, outperforms frontier models in generating correct kernels though it lacks in obtaining performant ones.

**Strengths:**

1. ConCuR leverages generated chain-of-thoughts (CoT) for better refinement of LLMs.
2. To avoid overthinking related issues, shortest reasoning lengths are selected out of datapoints with highest speedup.
3. To account for high speedup cases, authors also select data points with >5x speedup over baseline.

**Weaknesses:**

- Illustration in Figure 1 is not completely clear. I would encourage authors to refer other published papers for improving this.
- In section 3.3, it is not clear which model was used (under what conditions) to generate the data.
- In section 3.3, out of 90K total collected kernels, did authors find only 24K kernels to be correct?
- In section 3.5, only 4,892 samples are reported to be used for finetuning. It is not clear what happened to those 90K or 24K data points.
- In section 4.2 the definition of pass@k evaluation (lines 318, 319). Authors should refer to Chen et. la., 2021 human eval (https://arxiv.org/abs/2107.03374) paper for more clarity.
- The data curation pipeline does not inspire scientific innovation.
- Moreover, authors have not shown any way to determine the correctness of CoT steps. The bare assumption that generated CoTs are correct (even though the corresponding kernel maybe correct) isn't the right approach.
- Performance improvement of generated kernels is of vital importance to justify the cost spent in training/inference/generations. Lacking the details on hardware feedback processing does not inspire confidence in obtaining performant kernels.
- Case study shows naive details and does not show novel generated kernels that truly reflect thinking/reasoning from LLMs.
- Authors' approach achieves very low speedups in general. Other methods/approaches have demonstrated far more speedups in comparison.
- If authors' approach does not produce correct AND performant kernels more often then this approach does not contribute any innovation to the field.

**Questions:**

- How does this approach scale to low-resource languages such as Triton?
- Authors have not described in detail with evidence why is fast1 for L1 is worse than that of L2 in-spite of having easier problems in L1.
- Also refer to weakness section.

---

> ### Author Response · Authors · 2025-11-22
>
> Response to Reviewer `k3BA`: Thank you for your comments.
>
> ---
>
> > **W1:** *'
> Illustration in Figure 1 is not completely clear. I would encourage authors to refer other published papers for improving this.'*
>
> **A:** Thank you for your suggestion. We have revised our overview figure to make it clear. Moreover, could you please indicate which part of our figure is not clear and what papers could we refer to? Thank you.
>
> > **W2:** *'In section 3.3, it is not clear which model was used (under what conditions) to generate the data.'*
>
> **A:** Thank you for pointing this out. We have stated in our paper that we used Kevin-32B in **Section 3.3**, and the prompt we used in **Appendix C** to generate data. You can also find details in our **Reproducibility Statement**.
>
> > **W3:** *'In section 3.3, out of 90K total collected kernels, did authors find only 24K kernels to be correct?'*
>
> **A:** Thank you for this question. Yes, we did. This corresponds to the pass@1 accuracy of Kevin we reported. You can also refer to pass@1 accuracy of other models.
>
> > **W4:** *'In section 3.5, only 4,892 samples are reported to be used for finetuning. It is not clear what happened to those 90K or 24K data points.'*
>
> **A:** We have clearly described our data curation pipeline in **Section 3.5** in our original version. For your convenience, we refer our paper here:
>
> *Our dataset, ConCuR, consists of three parts.
> (a) For each task, we generated five kernels. If the kernel with the shortest reasoning length achieves the highest speedup, we add it to our dataset. We have 3,934 samples in this part.
> (b) We included kernels with speedups greater than 5, as these high-performance kernels are valuable for learning. This part contains 414 samples.
> (c) We need to balance the ratio of these two types of tasks in our dataset. To this end, we identified 544 samples with CUDA kernels for single operators and their CoTs. Based on these three parts, ConCuR, comprising 4,892 examples, considers speedup, reasoning quality, and task distribution.*
>
> > **W5:** *'In section 4.2 the definition of pass@k evaluation (lines 318, 319). Authors should refer to Chen et. la., 2021 human eval (https://arxiv.org/abs/2107.03374) paper for more clarity.'*
>
> **A:** Thank you for suggestion. We have added the citation.
>
> > **W6:** *'The data curation pipeline does not inspire scientific innovation.'*
>
> **A:** Thank you for feedback. However, we argue that constructing the first dataset of PyTorch and CUDA kernels generations with CoT is important since it supports works on kernel generation domain by addressing a significant concern that community lack of high-quality kernels. Based on our dataset, we successfully trained KernelCoder, which is a powerful model to generate kernels. In conclusion, we present our observation about the conciseness of CoT, release a powerful model, and propose a metric to quantify the difficulty of kernel generation problems, which we think will be beneficial to the community.
>
> > **W7:** *'Moreover, authors have not shown any way to determine the correctness of CoT steps. The bare assumption that generated CoTs are correct (even though the corresponding kernel maybe correct) isn't the right approach.'*
>
> **A:** We appreciate proposing this point. While verification of CoT steps could contribute to the robustness of generating kernels, they are beyond the scope of this work. Since it requires techniques from software engineering such as semantics proof or testing, which involve massive work.
>
> > **W8:** *'Performance improvement of generated kernels is of vital importance to justify the cost spent in training/inference/generations. Lacking the details on hardware feedback processing does not inspire confidence in obtaining performant kernels.'*
>
> **A:** We do not fully understand your point. Could you please explain what does it mean by *'inspire confidence'*? We regard you are indicating that we should optimize kernels with hardware feedback. This question is beyond our scope. To utilize hardware feedback, it may need test-scaling methods. However, our contribution is constructing the first PyTorch-CUDA dataset and a powerful base model. Moreoevr, KernelCoder, which is a lighter and faster model that can be easily served locally, it can be integrated into current multi-agent frameworks such as CudaForge (https://arxiv.org/abs/2511.01884) to obtain hardware feedback. We see this paper as a foundational work and leave this for future improvement plan as a key factor.
>
> > **W9:** *'Case study shows naive details and does not show novel generated kernels that truly reflect thinking/reasoning from LLMs.'*
>
> **A:** Thank you for pointing this out. We have added additional information and details in the revised version. The case study provides a comparison between concise, logical reasoning and prolonged, chaotic reasoning, serving as further evidence to support our observation.

---

> ### Author Response · Authors · 2025-11-22
>
> > **W10:** *'Authors' approach achieves very low speedups in general. Other methods/approaches have demonstrated far more speedups in comparison.'*
>
> **A:** Thank you for raising this concern. To our knowledge, we do not know 'methods that demonstrated far more speedups in comparsion', in forms of fine-tune models. Moreover, we reached competitive performance given limited computational resources.
>
> > **W11:** *'If authors' approach does not produce correct AND performant kernels more often then this approach does not contribute any innovation to the field.'*
>
> **A:** We do not know what is you meaning of 'innovation to field'. Our contribution of constructing the first dataset of PyTorch and CUDA kernels generations with CoT is important since it supports works on kernel generation domain. Moreover, we present our observation about the conciseness of CoT, release a powerful model, and propose a metric to quantify the difficulty of kernel generation problems. Furthermore, as stated in **W8**, our method is orthogonal to methods like test-time scaling, which can be also applied to further boost the performance of the generated kernels.  Overall, in our view, we do believe that we make contributions and innovations to the field.
>
> > **Q1:** *'How does this approach scale to low-resource languages such as Triton?'*
>
> **A:** Thank you for your question. We aim to contribute to CUDA kernel generation domain, which is a practical and influential domain. Although your question is out of our scope, we believe that our data collection and curation pipeline can also be used to construct dataset of PyTorch and Triton kernel.
>
> > **Q2:** *'Authors have not described in detail with evidence why is fast1 for L1 is worse than that of L2 in-spite of having easier problems in L1.'*
>
> **A:** Thank you for your comment, and we apologize for any confusion caused by our description. In the original KernelBench settings, Level 1 and Level 2 do not correspond to difficulty but rather to structural complexity of the operators/models (Level 1: single-kernel operators; Level 2: simple fusion patterns). According to our definition of "difficulty," which is measured by average reasoning length, Level 2 problems are not necessarily harder than Level 1. This was already stated in Section 6 of our original version. Moreover, we described two different optimization patterns for optimizing problems from KernelBench Levels 1 and 2 in Section 3.5 of the original version, which partially explains why "fast1 for L1 is worse than that of L2."
>
> Thank you for feedbacks again. While we acknowledge that our work may be exploratory, we believe some of your suggestions and concerns can be a foundation for our future work.

---

### Official Review · Reviewer_JCxH · 2025-11-02

**Soundness:** 2
**Presentation:** 2
**Contribution:** 3
**Rating:** 6
**Confidence:** 3

**Summary:**

This work presents ConCuR, an SFT curated dataset that pairs PyTorch code, reasoning traces, and corresponding CUDA kernels, claimed to be the first of its kind. Built through an automated synthetic pipeline, ConCuR enables training of KernelCoder, a model fine-tuned specifically for kernel generation. The dataset facilitates stronger reasoning-to-code alignment, leading to state-of-the-art performance on the KernelBench benchmark and outperforming both open-source and proprietary models.

**Strengths:**

* The work is well-motivated, addressing the high cost and expertise required for developing efficient GPU kernel data.
* The evaluation setup includes a reasonable number and balanced distribution of models, providing adequate coverage for assessing the proposed method’s effectiveness.
* The work presents interesting insights into how reasoning can enhance kernel code generation, highlighting the potential benefits of integrating structured reasoning traces into low-level code synthesis.

**Weaknesses:**

* Some sections of the manuscript, particularly those describing the data synthesis and curation pipeline, would benefit from language refinement and stylistic polishing to improve clarity and readability.
* The evaluation setup is somewhat limited, relying on a single fine-tuned model and one benchmark (KernelBench). Expanding the evaluation to include additional benchmarks, such as TritonBench, and more diverse fine-tuned models would strengthen the empirical validation and demonstrate broader applicability.
* While the related work section covers key benchmarks and datasets relevant to kernel generation, it lacks a comparative table or structured analysis that clearly contrasts the proposed benchmark with existing ones. Including such a comparison, highlighting differences in dataset size, performance, cost, and methodology, would make the contribution’s advantages more explicit and easier to assess.

**Questions:**

* Could the authors clarify how task difficulty was considered or quantified during the dataset construction? The manuscript discusses reasoning length as an indicator of difficulty but does not clearly explain how this factor influenced data curation (if it did). Clarifying the role of task difficulty and its potential impact on the reported results would help strengthen the connection between the dataset design and the discussion section.

---

> ### Author Response · Authors · 2025-11-21
>
> Response to Reviewer `JCxH`: Thank you so much for your insightful comments! We address your concerns point by point as follows:
>
> ---
>
> > **W1:** *"Some sections of the manuscript, particularly those describing the data synthesis and curation pipeline, would benefit from language refinement and stylistic polishing to improve clarity and readability."*
>
> **A:** We appreciate the suggestion and have revised the paper to enhance clarity and readability.
>
> > **W2:** *""The evaluation setup is somewhat limited, relying on a single fine-tuned model and one benchmark (KernelBench). Expanding the evaluation to include additional benchmarks, such as TritonBench, and more diverse fine-tuned models would strengthen the empirical validation and demonstrate broader applicability.*
>
> **A:** We totally understand your point. However, KernelBench is the only verfied and widely-used benchmark of **CUDA Kernel Generation**, since **TritonBench** is specifically designed for Triton kernels which is not applicapable to CUDA kernels. Additionally, we have implemented experiments on different base models to validate our dataset. The results demonstrated here is added to Section 5.2 in revised version.
>
> | Model | Level 1 Exec ↑ | Level 1 fast₁ ↑ | Level 2 Exec ↑ | Level 2 fast₁ ↑ |
> |---|---|---|---|---|
> | Qwen3-8B          | 31.0           | 12.0            | 53.0           | 32.0 |
> | **Qwen3-8B-SFT**  | 47.0           | 10.0            | 89.0           | 58.0 |
> | Qwen3-32B         | 68.0           | 21.0            | 82.0           | 37.0 |
> | **Qwen3-32B-SFT** | 72.0           | 21.0            | 94.0           | 68.0 |
> | QwQ-32B           | 55.0           | 12.0            | 76.0           | 46.0 |
> | **KernelCoder**   | 91.0           | 32.0            | 95.0           | 68.0 |
>
> We fine-tune three different base models on ConCuR. All fine-tuned models outperform their respective base models. This result further strengthens the empirical evidence and highlights the broader applicability of our ConCuR dataset.
>
> > **W3:** *"A comparative table or structured analysis that clearly contrasts the proposed benchmark with existing ones."*
>
> **A:** Thank you for this suggestion. We listed this table as follows and also add this table to section 4.3 in our paper.
>
> | Model |Compatibility |Params(B) | Post-training method | Number of training samples | Train epoches | Pass@10 on Level1/2 |
> |:---|:---|:---|:---|:---|:---|:---|
> |Kevin| CUDA| 32 | GRPO | 180* | NA | 86/90 |
> |AutoTriton|Triton| 8 | SFT+GRPO | 14102 for SFT, 6302 for GRPO | 3 epochs for sft + 1 epoch for GRPO (128+512 GPU hours on A800)| 68/88|
> |KernelLLM|Triton| 8 | SFT | ~25000 | 10 (192 GPU hours on H200) | 52/34 |
> |Deepseek-R1-0528|CUDA| 685 | NA | NA | NA | 90/97 |
> |**KernelCoder (ours)**|CUDA| 32 | SFT | 4892 | 3 (64 GPU hours on A100) | 91/95 |
>
> *: Kevin used 180 problems from KernelBench. For each problem, it explores 16 trajectories parallelly with 8 refinement steps.
>
> As shown in the table, KernelCoder achieved better performance with fewer number of training samples and computation cost.
>
> > **Q1:** *"Clarify how task difficulty was considered or quantified during the dataset construction."*
>
> **A:** Thank you for the insightful comment. Initially, we intended to consider "difficulty level" as a factor in curating the dataset. However, given the relatively small number of samples selected by our current method, we opted to include problems across all difficulty levels. To further investigate the impact of difficulty, we conducted an experiment using a subset of 2,210 samples deemed the most difficult according to our definition, out of a total of 4,892 samples. The results are summarized below:
> |Number of Data | Level1 (pass@10) | Level2 (pass@10) |
> |:---|:---|:---|
> |2210| 86.0 | 94.0 |
> |4982| 91.0 | 95.0 |
>
> The results indicate that the performance on the smaller, difficulty-biased subset is comparable to full dataset. This suggests that filtering based on difficulty does not significantly compromise model performance and supports the validity of our difficulty definition. To report the best overall performance, we therefore present ConCuR results using the full set of 4,892 samples.
>
> We truly appreciate the time and effort you’ve dedicated to reviewing our submission. We have improved our paper following your suggestions. We hope this can address your concerns.

---

> > ### Comment · Reviewer_JCxH · 2025-11-22
> >
> > > W1
> >
> > Great!
> >
> > > W2
> >
> > * Thanks for clarifying compatibility limitation regarding other benchmarks, I suggest to adde `cuda compatibility` to the benchmarks comparative table (added in response to W3).
> > * Thanks for addition additional experiments, any ideas why `Qwen3-8B-SFT` is performing worst than the baseline for Level 1 fast?
> >
> > > W3
> >
> > Great !
> >
> > > Q1
> >
> > Thanks for performing the ablation on difficulty. The `difficulty-biased` subset appears quite large, nearly half the full dataset. Is there a way to construct a more selective `super-difficulty-biased` subset? In addition, the result for `Level 1` does not seem directly comparable to the result obtained on the full set. Overall, I'm not fully convinced by the conclusions drawn from this ablation.

---

> ### Author Response · Authors · 2025-11-25
>
> Thanks for your response.
> > **W2:** *'The evaluation setup is somewhat limited, relying on a single fine-tuned model and one benchmark (KernelBench). Expanding the evaluation to include additional benchmarks, such as TritonBench, and more diverse fine-tuned models would strengthen the empirical validation and demonstrate broader applicability.'*
> * We have revised our table as per your suggestion.
> * It may stem from the limitation of the intrinsic ability of an 8B model. It may fail to learn the excellent patterns of reasoning and generate correct and fast kernels.
>
> > **Q1:** *'Clarify how task difficulty was considered or quantified during the dataset construction.'*
>
> As per your suggestion, we have further implemented the following ablation.
>
> |Number of Data | Level1 (pass@10) | Level2 (pass@10) |
> |---|---|---|
> |552 | 72.0 | 84.0|
> |1105 | 80.0 | 89.0|
> |2210 | 86.0 | 94.0 |
> |4982 | 91.0 | 95.0 |
>
> The results keep the previous trend, which is predictable. While filtering based on difficulty does not significantly compromise model performance, scaling is a dominant factor (which is our intuition that publishing the full dataset consists of 4892 samples).  Although learning from successful generations of hard problems is beneficial, it is also vital to learn from high-quality generations of simple problems. Additionally, can you provide any clues about which part you are *'not fully convinced'*? Thanks for your valuable feedback again.

---

### Official Review · Reviewer_QzMf · 2025-11-03

**Soundness:** 2
**Presentation:** 3
**Contribution:** 3
**Rating:** 2
**Confidence:** 3

**Summary:**

The paper proposes a dataset of reasoning traces for kernel generation that can be used as the basis of future SFT work on kernel llms.
It observes that for traces generated with current-generation LLMs, long traces correlate with _worse_ generation accuracy, and thus suggests to curate the dataset by selecting short traces that resulted in correct kernels.

**Strengths:**

* An open dataset like the one proposed would be valuable to the community
* The filtering based on trace length seems like an interesting approach.
* The reclassification of kernel bench tasks into more sensible difficulty classes is useful.

**Weaknesses:**

KernelBench is _not_ a reliable benchmark. Many of its shapes are too small, and the choice of pytorch eager as a baseline means you're mostly profiling against overhead.
To get a meaningful interpretation of the speed of the generated kernels, it would be important to calculate their speed-of-light, i.e., how long they would take to execute based on the GPUs flops and memory bandwidth, and then show what percentage of this speed is achieved.

Furthermore, I'm worried about about dataset contamination. The kernels in KernelBench were chosen because they are common operations, and I'd be surprised if among the 9,789 tasks you selected from kernel book, there was not a significant overlap.
This isn't a bad thing for the dataset to be released in itself (if I want to train a kernel llm, I do want all the popular kernels in the training set, after all), but it makes it hard to trust any of the metrics reported in the paper.

I would also say that the central observation about conciseness should be stated a bit more carefully. When generated by current-generation LLMs, short traces work better than long traces. That might not be the case for better future LLMs that don't "wait" so often, or for human-generated traces.

**Questions:**

(How) have you ensured that the tasks tested on in kernel bench are not part of the training set from kernel book.

Have you validated that the shapes for which speed-ups are reported are actually of a meaningfully large size. How do they compare against torch compile? What percentage of the speed-of-light is achieved?

For getting the fastest of 10 generations, do you select the time of the fastest run, or do you select the code of the fastest run, and re-time it again independently?

---

> ### Author Response · Authors · 2025-11-21
>
> Response to Reviewer `QzMf`: Thank you for your detailed feedback and valuable suggestions.
>
> ---
>
> >**W1:** *'KernelBench is not a reliable benchmark. Many of its shapes are too small, and the choice of pytorch eager as a baseline means you're mostly profiling against overhead. To get a meaningful interpretation of the speed of the generated kernels, it would be important to calculate their speed-of-light, i.e., how long they would take to execute based on the GPUs flops and memory bandwidth, and then show what percentage of this speed is achieved.'*, **Q2:** *'Have you validated that the shapes for which speed-ups are reported are actually of a meaningfully large size. How do they compare against torch compile? What percentage of the speed-of-light is achieved?'*
>
> **A:** Thank you for pointing this out. However, the issue you mentioned — *'many of its shapes are too small, and using PyTorch eager as a baseline means you are mostly profiling against overhead'* — **has already been addressed in the July release of KernelBench v0.1**. According to the official KernelBench v0.1 documentation (https://scalingintelligence.stanford.edu/blogs/kernelbenchv01/), the authors scaled up tensor dimensions and batch sizes to ensure that all Level 1 and Level 2 tests fall within an **ideal 1–15 ms execution window on an H100**, thereby significantly reducing overhead effects and allowing the benchmarks to reflect true speed-of-light performance characteristics. Since our experiments strictly followed KernelBench v0.1, this concern should have minimal impact on our reported results.

---

> ### Author Response · Authors · 2025-11-21
>
> > **W2:** *'Furthermore, I'm worried about about dataset contamination. The kernels in KernelBench were chosen because they are common operations, and I'd be surprised if among the 9,789 tasks you selected from kernel book, there was not a significant overlap. This isn't a bad thing for the dataset to be released in itself (if I want to train a kernel llm, I do want all the popular kernels in the training set, after all), but it makes it hard to trust any of the metrics reported in the paper.'*, **Q1:** *'(How) have you ensured that the tasks tested on in kernel bench are not part of the training set from kernel book.'*
>
> **A:** We deeply understand your point. While the risk of overlap between KernelBench test tasks and ConCuR training data does exist, we argue that **such overlap is largely acceptable and unavoidable** in the kernel generation domain.
>
> 1. **Kernel generation is inherently instance-specific:** even for a single operator or model with a fixed shape, discovering any new optimization (even a marginal 1.01× improvement) directly benefits existing frameworks, especially for widely used operators. At the same time, the space of valid and meaningful operators or model structures is fundamentally limited. The task cannot be “fully generalized,” as we cannot construct arbitrary operators without grounding them in real, practical computation patterns. This limitation is also evidenced by the construction of KernelBook: it collects all PyTorch modules for which unit tests can be generated, covering essentially the entire set of meaningful operators available in the community. Consequently, the total number of practically useful operators or model structures is well under 20,000, and any reasonable test set will inevitably overlap in operator type or structure with the training set. For these reasons, creating a completely disjoint (yet still meaningful) dataset for kernel generation is effectively impossible. Thus, some degree of overlap between training and test distributions is intrinsic to this domain.
>
> 2. **This "limitation", if any, is shared across prior papers.**
> It is important to emphasize that similar overlap exists in nearly all prior GPU kernel generation research. For example, Kevin (https://arxiv.org/pdf/2507.11948) and CUDA-L1 (https://arxiv.org/pdf/2507.14111) directly use the problems from KernelBench as part of their training data. Likewise, AutoTriton follows a data collection strategy that is conceptually aligned with KernelBook: it aggregates operators from real-world machine learning frameworks and does not attempt to create a strictly disjoint training–testing split.
> This consistency across prior methods reflects a common understanding in the community: given the limited pool of real, meaningful operators, strict separation between training and evaluation tasks is neither realistic nor representative of practical kernel optimization workloads.
>
> 3. **ConCuR is the first dataset to provide PyTorch–CUDA kernel pairs.**
> ConCuR is, to the best of our knowledge, the first dataset that systematically pairs PyTorch operators with their corresponding CUDA kernels. Because no prior dataset of this form exists, there is no established methodology or standard practice for ensuring strict disjointness from benchmarks like KernelBench. Furthermore, constructing such a dataset inherently requires covering widely used PyTorch operators—the same operators that KernelBench evaluates—making overlap even more difficult to avoid.
> In this sense, ConCuR breaks new ground and inevitably faces challenges that previous works, which relied directly on KernelBench or handcrafted small collections, did not need to address. Establishing principled methodologies for dataset construction and generalization evaluation is part of the broader future direction of this research area.

---

> ### Author Response · Authors · 2025-11-21
>
> > **W3:** *'I would also say that the central observation about conciseness should be stated a bit more carefully. When generated by current-generation LLMs, short traces work better than long traces. That might not be the case for better future LLMs that don't "wait" so often, or for human-generated traces.'*
>
> **A:** We appreciate this suggestion. Your assumption is certainly reasonable and we have revised our claim. Additionally, as you pointed out, “this might not be the case for better future LLMs.”, our observation offers a unique perspective: it suggests that as future LLMs become more capable of reasoning more logically and efficiently, so that the benefit of longer reasoning traces may become more pronounced, rather than diminished. In this sense, our finding not only aligns with your speculation but also provides empirical support for the potential of future models to utilize longer, more coherent reasoning sequences effectively. We hope that this observation will contribute not only to the kernel generation domain but also to the broader development of general-purpose foundation models.
>
>
> > **Q3:** *'For getting the fastest of 10 generations, do you select the time of the fastest run, or do you select the code of the fastest run, and re-time it again independently?'*
>
> **A:** Thank you for this question. we selected the time of the fastest run, since we followed the regular pipeline of KernelBench to measure the performance of generated kernel. The kernel running time is by default an average of 100 tests after 3 warm-ups.
>
> We truly appreciate the time and effort you’ve dedicated to reviewing our submission and your insightful feedbacks. We agree that kernel generation is an emerging domain and needs more standard benchmarks and methods. We believe our statement can address your concerns effectively.

---

### Author Response · Authors · 2025-11-21
**Summary of reviews**

Thanks to all reviewers for providing insightful feedback and suggestions. We conclude reviews and respond to the common concerns or questions here.

To sum up, we are encouraged to find all reviewers agree that:

>1. Our paper shows reasonable and strong motivation, addressing the high cost and expertise required for developing efficient GPU kernel data. (Reviewer JCxH)
2. Our *'insightful observation that concise reasoning traces lead to better kernel generation performance'* (Reviewer 2Vjz), *'is an interesting approach'* (Reviewer QzMf), and helps the construction of the dataset, ConCuR (Reviewer k3BA).
3. Our *'open dataset like the one proposed would be valuable to the community'* (Reviewer QzMf) and our model KernelCoder, '*demonstrates strong technical quality. It is a state-of-the-art model capable of generating correct and efficient CUDA kernels*' (Reviewer 2Vjz).
4. *'The inclusion of an ablation study further strengthens the work by demonstrating the effectiveness of the dataset curation pipeline.'*

Moreover, reviewers raise a common concern, and we address them here:

> Requesting more information about our training/inference/test configurations. (Reviewer QzMf, Reviewer JCxH, Reviewer k3BA, Reviewer 2Vjz)

**A:** We have added information and details as per request, to make our paper more reproducible. Overall, we have added the following content to our revised paper.
1. We revised the overview figure to improve clarity. (Reviewer k3BA)
2. We carefully revised the description of the data curation pipeline in Section 3 to ensure accuracy (Reviewer QzMf) as well as clarity and readability (Reviewer JCxH).
3. We added Section 4.3 and Table 3, which compare the efficiency of training samples and computational resources, highlighting the efficiency of our dataset and model. (Reviewer JCxH)
4. We added Section 5.2 and Table 5, reporting Pass@10 results for different base models and their fine-tuned versions, thereby strengthening empirical evidence and demonstrating the broader applicability of our ConCuR dataset. (Reviewer JCxH)
5. We revised Table 7 to include the results of DeepSeek-R1-0528. (Reviewer 2Vjz)

Additionally, we noticed that Reviewer `k3BA` has proposed 11 weaknesses, but 3 of them are just questions (W2-W4) which **we have clearly claimed in our original paper**.

To conclude, the domain of (CUDA) kernel generation is still emerging, and the community will undoubtedly benefit from more comprehensive benchmarks and standardized evaluation protocols. This paper represents an early exploratory effort, and we believe it lays an important foundation for future research and the broader automation of kernel generation. We once again thank all reviewers for their time and insightful feedback.

---

### Meta-Review · Area_Chair_Md3G · 2026-01-09

**Summary:**

With ratings 2266, this is quite borderline. Reviewers appreciated the data contribution and the performance as well as the central insight of conciseness is better. The authors partly addressed the important issues of evaluation validity and overlap, but it's unclear if the reviewer is convinced. Looking into these points,  the authors claim that the validity is fixed in later versions but admits that overlap is unavoidable. The reviews are not too informative on the key claim that conciseness is the most important. A bit of my own review reveals that the author made an arbitrary division of difficulty of KernelBench by binning the reasoning length and claimed it to be a contribution. Given the lack of enthusiasm from the reviewers, that the topic is quite niche and the contributions are a bit diffuse, voting reject.

## Strengths
* Dataset contribution (QzMf, JCxH, 2Vjz): Open dataset valuable to community; well-motivated addressing high GPU kernel development cost
* Concise reasoning insight (QzMf, k3BA, 2Vjz): Short reasoning traces outperform long ones; CoT filtering based on trace length
* Strong model performance (2Vjz): KernelCoder achieves state-of-the-art on KernelBench L1/L2, generates correct efficient CUDA kernels
* Good experimental setup (JCxH, 2Vjz): Balanced model coverage; includes ablation studies validating curation pipeline

## Weaknesses
* KernelBench unreliable/limited evaluation (QzMf, JCxH, k3BA): Shapes too small, baseline measures overhead not true performance; single benchmark insufficient;
* Dataset contamination concerns (QzMf, k3BA): Likely overlap between 9,789 selected kernels and common KernelBench operations; correctness pipeline unclear (24K→4,892 samples?)
* Low speedups & missing performance details (k3BA, 2Vjz): Very low speedups compared to other methods; lacks hardware feedback processing; no speed-of-light calculations; 5× threshold unexplained
* Unclear methodology (k3BA, 2Vjz): Model used for data generation unclear; correctness analysis setup missing (random inputs, tolerances); Kevin model choice unjustified vs DeepSeek-R1
* No CoT correctness validation (k3BA): No way to verify CoT steps are correct even if kernel works; bare assumption insufficient

Minor: ConCuR is maybe not the best name. High chance of overlap, and more directily invokes concurrency instead of Concise Cuda Reasoning.

**Reviewer Concerns:**

only partly addressed, I think likely not too convincingly.

**Reviewer Scores:**

not sure

---

### Decision · Program_Chairs · 2026-01-26

Reject